# High Resolution and Fast Face Completion via Progressively Attentive GANs

## Abstract

Face completion is a challenging task with the difficulty level increasing significantly with respect to high resolution, the complexity of "holes" and the controllable attributes of filled-in fragments. Our system addresses the challenges by learning a fully end-to-end framework that trains generative adversarial networks (GANs) progressively from low resolution to high resolution with conditional vectors encoding controllable attributes. We design a novel coarse-to-fine attentive module network architecture. Our model is encouraged to attend on finer details while the network is growing to a higher resolution, thus being capable of showing progressive attention to different frequency components in a coarse-to-fine way. We term the module **Frequency-oriented Attentive Module** (FAM). Our system can complete faces with large structural and appearance variations using a single feed-forward pass of computation with mean inference time of 0.54 seconds for images at $1024 \times 1024$ resolution. A pilot human study shows our approach outperforms state-of-the-art face completion methods. The code will be released upon publication.

## 1 Introduction

Image completion is a technique to replace target regions, either missing or unwanted, of images with synthetic content so that the completed images look natural, realistic and appealing. Two types of methods have been used: data similarity driven methods and data distribution based generative methods. In the first paradigm, texture synthesis or patch matching are usually used (Efros & Leung, 1999; Kwatra et al., 2003; Criminisi et al., 2003; Wilczkowiak et al., 2005; Komodakis, 2006; Barnes et al., 2009; Darabi et al., 2012; Huang et al., 2014; Wexler et al., 2007). The second paradigm learns the underlying distribution governing the data generation with respect to the context and is able to synthesize novel content. Much progress (Iizuka et al., 2017; Yeh et al., 2017; Li et al., 2017; Yang et al., 2016; Denton et al., 2016; Pathak et al., 2016; Yu et al., 2018; Liu et al., 2018) has been made since the generative adversarial network (GAN) was proposed (Goodfellow et al., 2014).

We adopt the data distribution based generative method and focus on human face completion in this paper. Three important issues are addressed. *First*, previous methods are only able to complete faces at low resolutions (e.g. $176 \times 216$ (Iizuka et al., 2017) and $256 \times 256$ (Yu et al., 2018)). *Second*, most approaches cannot control the attributes of the synthesized content. Previous methods focus on generating random realistic content. However, users may want to complete the missing parts with certain properties (e.g. expressions). *Third*, most existing approaches (Iizuka et al., 2017; Yeh et al., 2017; Li et al., 2017) require post processing (e.g. Poisson Blending (Pérez et al., 2003)) or complex inference process (e.g. thousands of optimization iterations (Yeh et al., 2017) or repeatedly feeding an incomplete image to CNNs at multiple scales (Yang et al., 2016)).

To overcome the above limitations, we propose a novel progressively attentive GAN to complete face images at high resolution with multiple controllable attributes in a single forward pass without any post processing. We utilize facial landmarks as backbone guidance of face structures and propose a straightforward method of integrating them in our system. The training methodology of growing GANs progressively (Karras et al., 2017) is used to generate high-resolution images end-to-end. To avoid distorting the learned coarse structures when the network is growing to a higher resolution, we design a novel **Frequency-oriented Attentive Module (FAM)** to encourage the model to attend on finer details (i.e. higher-frequency structures, see Figure 1). A conditional version of our network is designed so that the appearance properties (e.g. male or female), and facial expressions of the synthesized faces can be controlled. Moreover, we design a set of loss functions inducing the network

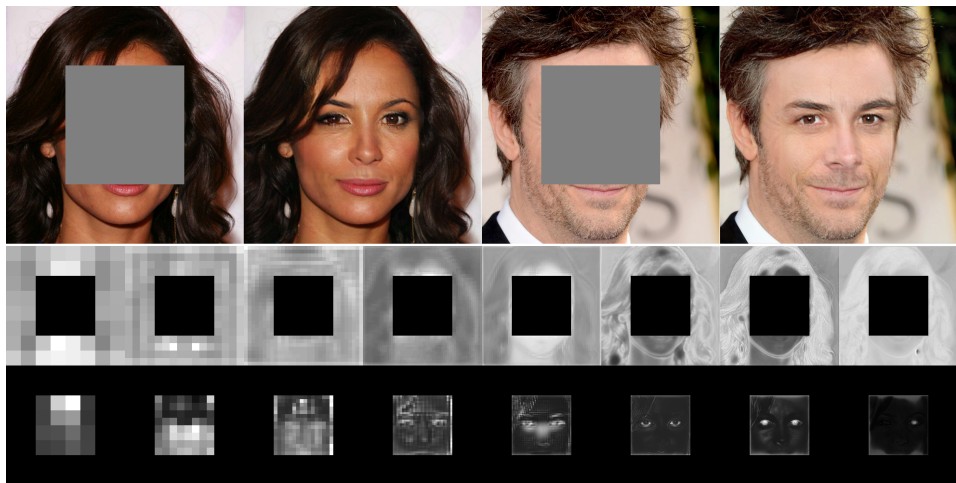

Figure 1: Face completion results of our method on CelebA-HQ (Karras et al., 2017). *Top*: our approach can complete face images at high resolution ($1024 \times 1024$). *Middle and Bottom*: the frequency-attention filters of readers and writers of the top left image. While the resolution increases from $8 \times 8$ to $1024 \times 1024$, the model attends on higher frequency information. Regions with rich details (e.g. eyes) get more attention, especially at high resolutions. Best viewed in magnification.

to blend the synthesized content with the contexts in a realistic way. Our method was compared with state-of-the-art approaches on a high-resolution face dataset CelebA-HQ (Karras et al., 2017). Both the evaluations and a pilot user study showed that our approach completed face images significantly more naturally than existing methods.

The main contributions of this paper are: (i) We propose a progressively attentive GAN architecture, which incorporates a novel frequency-oriented attention mechanism, to complete face images with random masks in much higher resolution than existing methods. (ii) A conditional version of our model is designed to control multiple attributes of the synthesized content. (iii) Our framework is able to complete images in a single forward pass, without any post-processing, and thus fast.

## 2 RELATED WORK

There is a large body of image completion literature. Early non-learning based algorithms (Efros & Leung, 1999; Bertalmio et al., 2000; 2003) complete missing content by propagating information from known neighborhoods, based on low level cues or global statistics (Levin et al., 2003). Texture synthesis and patch matching based approaches (Efros & Leung, 1999; Kwatra et al., 2003; Criminisi et al., 2003; Wilczkowiak et al., 2005; Komodakis, 2006; Barnes et al., 2009; Darabi et al., 2012; Huang et al., 2014; Wexler et al., 2007) find similar structures from the context of the input image or from an external database (Hays & Efros, 2007) and then paste them to fill in the holes.

Recent learning based methods have shown the capability of CNNs to complete large missing content. Based on existing GANs, the Context Encoder (CE) (Pathak et al., 2016) encodes the contexts of masked images to latent representations, and then decodes them to natural content images, which are pasted into the original contexts for completion. However, the synthesized content of CE is often blurry and has inconsistent boundaries. Given a trained generative model, Yeh et al. (Yeh et al., 2017) propose a framework to find the most plausible latent representations of contexts to complete masked images. The Generative Face Completion model (GFC) (Li et al., 2017) and the Global and Local Consistent model (GL) (Iizuka et al., 2017) use both global and local discriminators, combined with post processing, to complete images more coherently. Built on GL, Yu et al. (Yu et al., 2018) design a contextual attention layer (CTX) to help the model borrow contextual information from distant locations. Liu et al. (Liu et al., 2018) incorporates partial convolutions to handle irregular masks. Unfortunately, these approaches can only complete face images in relatively low resolutions (e.g. 176 x 216 (Iizuka et al., 2017) and $256 \times 256$ (Yu et al., 2018)). Yang et al. (Yang et al., 2016) combine a global content network and a texture network, and the networks are trained at multiple scales repeatedly to complete high-resolution images ($512 \times 512$). Like the patch matching based approaches, Yang et al. assume that the missing content always shares some similar textures with the context, which is improbable for the face completion task.

Many methods try to generate high quality images and stabilize the training process. The Laplacian GAN (Denton et al., 2015) uses a a cascade of CNNs to synthesize images from coarse to fine by generating high frequency information at different layers. The Deep Recurrent Attentive Writer (DRAW) architecture (Gregor et al., 2015), which is a spatial attention mechanism, generates images iteratively by learning to read and write parts of images at each time-step with a sequence of variational auto-encoders (VAEs). Unfortunately, these techniques are unable to synthesize high-resolution images (e.g. $64 \times 64$ (Gregor et al., 2015; Denton et al., 2015)). Karras et al. (Karras et al., 2017) put forward a progressive training methodology (Progressive GAN) to grow GANs from low to high resolution, and are able to generate realistic $1024 \times 1024$ images. However, since all the parameters remain trainable at the growing stage, learned coarse structures can be altered and distorted, and thus the training process of Progressive GANs is usually unstable. *Note that these generative models cannot be applied to the image completion task directly because they aim at generating natural content which are not necessarily consistent with the image contexts.*

## 3 APPROACH

### 3.1 PROBLEM FORMULATION

Denoted by $\Lambda$ an image lattice (e.g., $1024 \times 1024$ pixels). Let $I_\Lambda$ be an RGB image defined on the lattice $\Lambda$. Denote by $\Lambda_t$ and $\Lambda_c$ the target region to complete and the remaining context region respectively which form a partition of the lattice. Without loss of generality, we assume $\Lambda_t$ is a single connected component region. $I_{\Lambda_t}$ is masked out with the same gray pixel value. Let $M_\Lambda$ be a binary mask image with all pixels in $M_{\Lambda_t}$ being 1 and all pixels in $M_{\Lambda_c}$ being 0. For notational simplicity, we will omit the subscripts $\Lambda$, $\Lambda_t$ and $\Lambda_c$ when the text context is clear.

The objective of image completion is to generate a synthesized image $I^{syn}$ that looks natural, realistic and appealing for an observed image $I^{obs}$ with the target region $I^{obs}_{\Lambda_t}$ masked out in the ground-truth full image $I^{gt}$. Furthermore, it is desirable to control the completion according to a set of attributes which are assumed to be independent from each other (e.g., to respect the underlying intrinsic ambiguities due to the loss of information in the target region). Let $A = (a_1, \cdots, a_N)$ be a $N$-dim vector with $a_i \in \{0, 1\}$ encoding if a corresponding attribute appears ($a_i = 1$) or not ($a_i = 0$) (e.g. the "Male" attribute in Figure 5). We define the generator as,

$$I^{syn} = G(I^{obs}, M, A; \theta_G), \text{ subject to } I^{syn}_{\Lambda_c} \approx I^{obs}_{\Lambda_c} \tag{1}$$

where $\theta_G$ collects all parameters of the generator, and $\approx$ represents the two context regions, $I^{syn}_{\Lambda_c}$ and $I^{obs}_{\Lambda_c}$, need to kept very similar (both to be elaborated later).

### 3.2 THE PROPOSED METHOD

The proposed method is built on progressive GANs with well-executed combination between the network architecture, appropriate receipt of loss functions and a novel FAM.

Denote by $G_r$ and $D_r$ the generator and discriminator at resolution $r$ respectively, where $r \in \{1, \cdots, R\}$ is the index of resolution (e.g., $r = 1$ represents $4 \times 4$ and $r = R = 9$ represents $1024 \times 1024$). Accordingly, we have the observed corrupted image, its corresponding binary mask and its ground-truth uncorrupted image (in training only), $I^{obs}_r$, $M_r$ and $I^{gt}_r$ at each resolution.

*The generator*, $G_r$ takes as input the observed data $X^G_r$ and the attribute vector $A$, then outputs a completed image $I^{syn}_r$. It consists of two components,

$$I^{syn}_r = G_r(X^G_r, A; \theta_{G_r}) = G^{compl}_r(G^{enc}_r(X^G_r; \theta_{G^{enc}_r}), A; \theta_{G^{compl}_r}), \tag{2}$$

where $G^{enc}_r(\cdot)$ encodes the input $X^G_r$ to a latent low dimensional vector. The latent vector is concatenated with the attribute vector. The concatenated vector plays the role of the noise random variable $z$ in the original GAN. Then, $G^{compl}_r(\cdot)$ transforms the concatenated vector to a sample $I^{syn}_r$ (i.e., the completed image). $G^{enc}_r$ and $G^{compl}_r$ are mirrored to each other in the form of U-shape networks (Ronneberger et al., 2015; Newell et al., 2016). $\theta_{G_r} = (\theta_{G^{enc}_r}, \theta_{G^{compl}_r})$.

*The discriminator*, $D_r$ classifies its input $X^D_r$ and has two output branches, the fake versus real classification branch and the attribute prediction branch. It consists of three components: a shared feature backbone and two head classifiers. We have,

$$D_r(X^D_r; \theta_{D_r}) = \{D^{cls}_r(\mathbb{F}_r(X^D_r; \theta_{\mathbb{F}_r}); \theta_{D^{cls}_r}), D^{attr}_r(\mathbb{F}_r(X^D_r; \theta_{\mathbb{F}_r}); \theta_{D^{attr}_r})\} \tag{3}$$

Where the feature backbone, $\mathbb{F}_r(X_r^D; \theta_{\mathbb{F}_r})$ computes the feature map. On top of the feature map, the first head classifier, $D_{cls}(\cdot)$ computes binary classification between real and fake, and the second one, $D_{attr}(\cdot)$ predicts an attribute vector. All the parameters of the discriminator are collected by $\theta_{D_r} = (\theta_{\mathbb{F}_r}, \theta_{D_r^{cls}}, \theta_{D_r^{attr}})$. We note that the discriminator is only needed in training. We will omit the notations for parameters in equations when no confusion is caused.

*Progressive Growing of Generators and Discriminators.* Following the methodology proposed in (Karras et al., 2017), we start with training $G_1$ and $D_1$. Then, at the growing stage $r$ $(r > 1)$, $G_r = (G_{r-1}, G_r^{fade-in})$ is first created on top of the previous trained stage $G_{r-1}$ with newly added layers $G_r^{fade-in}$, and both of them are trainable in a fade-in process for smooth transition. The discriminators grow in the same way. **The alternation of $G_{r-1}$ in training stage $G_r$ may lead to the instability of the growing process and the loss of learned structures of the underlying image distribution, which motivates the proposed FAM in growing generators (Section 3.2.1).**

*Inputs for Generators and Discriminators, $X_r^G$ and $X_r^D$.* We have,

$$X_r^G = (\hat{I}_r^{obs}, M_r, L_r) \quad and \quad X_r^D = (I_r, L_r) \tag{4}$$

where $L_r$ represents the facial landmarks. Recent works (Isola et al., 2016; Wang et al., 2017; Zhu et al., 2017; Sangkloy et al., 2017; Xian et al., 2017; Chen & Hays, 2018) have shown the capability of GANs to translate sketches or edges to photo-realistic images. Given a corrupted image, it is better if the model is able to "draw" a sketch of face first, which provides a backbone guidance for image completion. We utilize the following methods to compute landmarks (Figure 2).

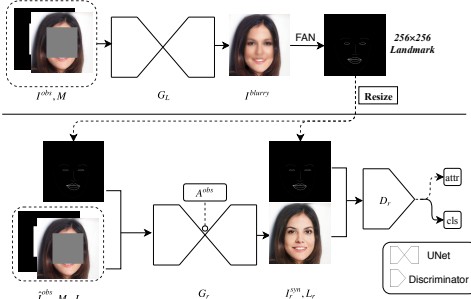

- In training, we extract landmarks from the un-corrupted image at $256 \times 256$ resolution using an off-the-shelf pre-trained Face Alignment Network (FAN) (Bulat & Tzimiropoulos, 2017) which achieved very good results for faces in the wild.

- In testing, to predict landmarks from corrupted images, we first train a single stage face completion network at $256 \times 256$ resolution, denoted by $G_L$, using reconstruction loss (Section 3.2.2) only. For a testing image, we use $G_L$ to generate a blurry completed image from which the landmarks are extracted with FAN.

Figure 2: Overview of our method. See text for details.

$L$ is up-sampled or down-sampled to $L_r$ to match the size of networks at different resolutions. $I_r$ in $X_r^D$ represents either the uncorrupted image or the image synthesized by $G_r$. For $\hat{I}_r^{obs}$ in $X_r^G$, we have $\hat{I}_1^{obs} = I_1^{obs}$, and $\hat{I}_r^{obs}$ is the output of FAM $(r > 1)$ as described in following section.

### 3.2.1    FAM: THE PROPOSED FREQUENCY-ORIENTED ATTENTIVE MODULE

Figure 3 illustrates the proposed FAM. To obtain a smooth transition from low to high resolutions, we design a FAM architecture (i.e. the red components in Figure 3), which is a frequency-oriented attention mechanism integrated along with the resolution change, so that the model learns filters to encourage $G_r^{fade-in}$ (blue components in Figure 3) to *read and write* information that are important at level $r$, but have not been handled well at level $r - 1$. By doing those,

- Our model attends on higher frequency signals as resolution increases (see Figure 1), thus improving the completion performance.

- Our model preserves what have been learned in previous progressive stages, thus furthering the stability of progressive GANs.

Existing approaches (Gregor et al., 2015; Yu et al., 2018) use spatial attention mechanisms to encourage networks to attend to selected parts of images (e.g. rectangular regions), while FAM is an attention model in the frequency domain. But different from a regular band-pass filter, the filters generated by FAM is predicted based on semantics of images which are enforced by the objective function (Eqn. 10), and thus is also sensitive to locations inferred on-the-fly in a coarse-to-fine manner. For instance, the model pays more attention to eye regions where the rich details aggregate, especially at a high resolution.

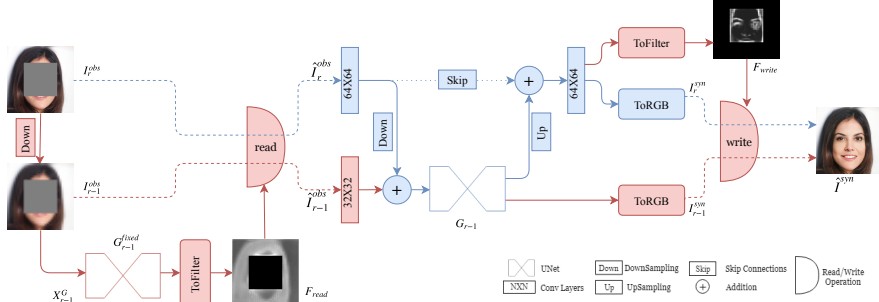

Figure 3: The proposed FAM used in growing GANs progressively. Here, we show the example of increasing resolutions from $32 \times 32$ to $64 \times 64$. See text for details. Best viewed in color and magnification.

Recall that at the growing stage $r$ ($r > 1$) in training, we have $G_r = (G_{r-1}, G_r^{fade-in})$. In the vanilla progressive GANs, we will compute the completed image $I_r^{syn} = G_r(I_r^{obs}, M_r, L_r, A)$ (simplified from Eqn. 2 for clarity). The previously trained $G_{r-1}$ can be changed while optimizing the stage $r$, which may lead to unexpected updating. Our FAM prevents $G_{r-1}$ from changing to wrong directions. To that end, we first introduce a *read* module which utilizes a *read* filter $\hat{F}_{read}$ to extract the most valuable information in both $I_r^{obs}$ and $I_{r-1}^{obs}$,

$$\hat{I}_r^{obs}, \hat{I}_{r-1}^{obs} = read(I_r^{obs}, I_{r-1}^{obs}, \hat{F}_{read}), \tag{5}$$

which is implemented by,

$$\hat{I}_r^{obs} = \hat{F}_{read} \odot (1 - M_r) \odot I_r^{obs}, \tag{6}$$

$$\hat{I}_{r-1}^{obs} = Downsample((1 - \hat{F}_{read}) \odot (1 - M_r) \odot \tilde{I}_{r-1}^{obs}), \tag{7}$$

where $\odot$ denotes element-wise multiplication. $\tilde{I}_{r-1}^{obs}$ is up-sampled from $I_{r-1}^{obs}$ to match the resolution of $I_r^{obs}$. $\tilde{I}_{r-1}^{obs}$ represents the blurred (i.e. low-frequency) version of $I_r^{obs}$ since high-frequency information is lost when $I_{r-1}^{obs}$ is down-sampled from $I_r^{obs}$. The *read* filter $\hat{F}_{read}$ is defined by,

$$\hat{F}_{read} = \beta \cdot F_{read} + \gamma, \qquad \beta : \left\{ \begin{array}{l} 2\alpha, \\ 2 - 2\alpha, \end{array} \right. \gamma : \left\{ \begin{array}{ll} 0, & \alpha \leq 0.5 \\ 2\alpha - 1, & 0.5 < \alpha \leq 1.0 \end{array} \right. \tag{8}$$

where $F_{read}$ is computed by $F_{read} = ToFilter(G_{r-1}^{fixed}(I_{r-1}^{obs}, M_{r-1}, L_{r-1}))$ using a trained generator $G_{r-1}^{fixed}$ with fixed weights and a small trainable network *ToFilter*. $\alpha$ is a weight increasing linearly from 0 to 1 proportional to the number of seen images during growing. $\hat{F}_{read}$ starts as an all-zero filter, is adjusted by a trainable *ToFilter* at the growing stages and eventually increased to all ones.

As illustrated in Figure 3, given the outputs from the *read* module, $\hat{I}_r^{obs}$ and $\hat{I}_{r-1}^{obs}$, we can generate the corresponding completed images, $I_r^{syn}$ and the up-sampled $I_{r-1}^{syn}$ respectively, and the *write* filter $F_{write}$. Then, to generate the final completed image at stage $r$, we also introduce a *write* module,

$$\hat{I}_r^{syn} = write(I_r^{syn}, I_{r-1}^{syn}, \hat{F}_{write}) \tag{9}$$

$$= (I_r^{syn} \cdot \alpha + I_{r-1}^{syn} \cdot (1 - \alpha)) \odot (1 - M_r) + (\hat{F}_{write} \odot I_r^{syn} + (1 - \hat{F}_{write}) \odot I_{r-1}^{syn}) \odot M_r,$$

where $\hat{F}_{write} = \beta \cdot F_{write} + \gamma$ and $F_{write}$ is predicted from the last feature maps. At a low resolution, both the *read* and *write* module exploit more on the low-frequency information, but gradually move to exploit higher-frequency information when resolution increases under the guidance of minimizing the objective function (Eqn. 10). $F_{read}$ and $F_{write}$ can be discarded when the growing process is done. A testing image only needs to go through one generator that is independent of FAM (blue flow in Figure 3). This is more efficient than the Laplacian GAN (Denton et al., 2015) that requires feeding a testing sample to a cascade of generators, and uses multiple discriminators in training.

### 3.2.2 LOSS FUNCTIONS

Beside extending the original adversarial loss function, we design three new loss functions to enforce sharp image completion.

**Adversarial Loss** Given an uncorrupted image $I^{gt}$, its attribute vector $A$, a mask $M$, landmarks $L$, and the corresponding corrupted image $I^{obs}$ we define the loss by, $l_{adv}(I^{gt}, M, L, I^{obs}, A|G, D) = \log(1 - D_{cls}(I^{syn}, L) + \log D_{cls}(I^{gt}, L)$, where $I^{syn} = G(I^{obs}, M, L, A)$.

128x128  256x256  512x512  1024x1024

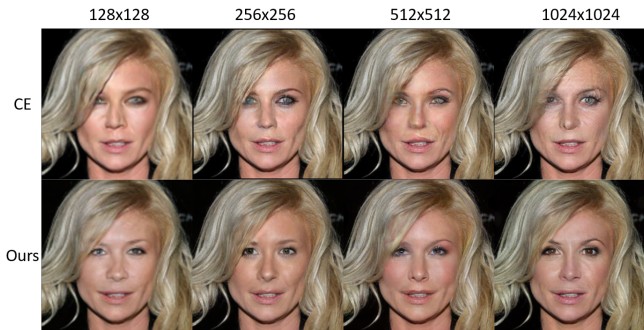 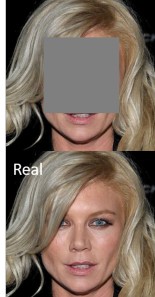

Figure 4: Comparison with Context Encoder (CE) on high-resolution face completion. While increasing resolution, CE generated more distorted images while our method produced sharper faces with more details.

**Attribute Loss** Similar to the InfoGAN models (Chen et al., 2016; Choi et al., 2017), for the attribute prediction head classifier in the discriminator, we define the attribute loss based on cross-entropy between the predicted attribute vector, $\hat{A}^{real} = D_{attr}(I^{real}, L)$ and $\hat{A}^{obs} = D_{attr}(I^{obs}, L)$ and the corresponding targets, $A$ for both a real uncorrupted image and a synthesized image. We have, $l_{attr}(I^{gt}, A, M, I^{obs}|G, D) = CrossEntropy(A, \hat{A}^{gt}) + CrossEntropy(A, \hat{A}^{obs})$.

**Reconstruction Loss** Since our method generates the entire completed face rather than only the target region, we define a wighted reconstruction loss $l_{rec}$ to preserve both the target region and the context region, which is defined as, $l_{rec}(I^{gt}, M, L, I^{obs}, A|G) = \|\kappa \odot M \odot I^{diff}\|_1 + \|(1 - \kappa) \odot (1 - M) \odot I^{diff}\|_1$, where $I^{diff} = I^{gt} - I^{syn}$ and $\kappa$ is the trade-off parameter.

**Feature Loss** In additional to the reconstruction loss in terms of pixel values, we also encourage the synthesized image to have a similar feature representation (Johnson et al., 2016) based on a pre-trained deep neural network $\phi$. Let $\phi_j$ be the activations of the $jth$ layer of $\phi$, the feature loss is defined by, $l_{feat}(I^{gt}, M, L, I^{obs}, A|\phi, G) = \|\phi_j(I^{gt}) - \phi_j(I^{syn}))\|_2^2$. In our experiments, $\phi_j$ is the $relu2\_2$ layer of a 16-layer VGG network (Simonyan & Zisserman, 2014) pre-trained on the ImageNet dataset (Russakovsky et al., 2015).

**Boundary Loss** To make the generator learn to blend the synthesized target region with the original context region, we further define a close-up reconstruction loss along the boundary of the mask. Similar to (Yeh et al., 2017), we first create a weighted kernel $w$ based on the mask image $M$. $w$ is computed by blurring the mask boundary in $M$ with a mean filter so that the pixels closer to the mask boundary are assigned larger weights. The kernel size of the mean filters is seven in our experiments. We have, $l_{bdy}(I^{gt}, M, L, I^{obs}, A|G) = \|w \odot (I^{gt} - I^{syn})\|_1$.

Our model is trained end-to-end by integrating the expected loss of the different loss functions defined above under the minimax game setting. We have,

$$\min_G \max_D \mathcal{L}_{adv}(G, D) + \lambda_1 \mathcal{L}_{attr}(G, D) + \lambda_2 \mathcal{L}_{rec}(G) + \lambda_3 \mathcal{L}_{feat}(G, \phi) + \lambda_4 \mathcal{L}_{bdy}(G). \quad (10)$$

Where $\lambda_i$'s are trade-off parameters between different loss terms.

**Training without Multiple Controllable Attributes.** To that end, since it is a special case of the proposed formulation stated above, we can simply remove the components involving attributes such as the attribute loss in a straightforward way. The resulting system still enjoys end-to-end learning.

## 4   EXPERIMENTS

**Datasets and Experiment Settings** We used the CelebA-HQ (Karras et al., 2017) dataset for evaluation. It contains 30,000 aligned face images at $1024 \times 1024$ resolution. The dataset is split randomly **while ensuring there is no identity overlap between test/training sets**: 3,009 images for testing, and 26,991 for training. There were two types of masks: center and random. The center mask was a square region in the middle of the image with a side length of half the size of the image. The random masks, generated in a similar way to previous methods (Iizuka et al., 2017; Yu et al., 2018), were rectangular regions with random width-to-height ratios, sizes and locations covering about $5\%$ to $25\%$ of the original images. Hyper-parameters used for training are listed in the supplemental materials.

**Quality Comparison with the Context Encoder** Our method was compared with the Context Encoder (CE) (Pathak et al., 2016) on high-resolution face completion. Since the original networks

of CE were designed for $128 \times 128$ images, we used a naive approach to fit it to different resolutions. One, two, and three convolutional layers were added to the encoder, decoder and discriminator for $256 \times 256$, $512 \times 512$ and $1024 \times 1024$ networks respectively. The result (Figure 4) shows that, when the resolution increased, our method learned details incrementally and synthesized sharper faces, while CE generated poorer images with more distortions.

Table 1: The quantitative comparison between our method and state-of-the-art methods

| Method | Resolution | L1 (%) | L2 (%) | PSNR |
|---|---|---|---|---|
| GL (Iizuka et al., 2017) | $128 \times 128$ | 9.34 | 1.75 | 18.22 |
| Ours | $128 \times 128$ | **7.8** | **1.42** | **19.15** |
| CTX (Yu et al., 2018) | $256 \times 256$ | 8.53 | 1.75 | 18.41 |
| Ours | $256 \times 256$ | **7.05** | **1.21** | **19.97** |

**Quantitative Comparison with State-of-the-art Methods** As noted in literatures (Yeh et al., 2017; Yu et al., 2018), reconstruction metrics such as mean $L1$, $L2$ errors and peak signal-to-noise ratio (RSNR) that are commonly used are not good quantitative evaluation metrics for inpainting methods since image completion aims at completing missing regions with plausible content rather than reconstructing it. As a reference, we show the comparison between our method and state-of-the-art models at their reported resolutions respectively: $128 \times 128$ for GL with center masks (using implementation of (Yu et al., 2018)) and $256 \times 256$ for CTX with random masks (Table 1).

**Semantic Completion** We first trained a high-resolution ($1024 \times 1024$) model with center masks (examples shown in Figure 5) to test whether our model is capable of learning high-level semantics and structures of faces and synthesize large missing regions. The second model was trained with random masks, but was able to handle arbitrary (e.g. irregular hand-drawn) shapes of masks. The result (Figure 5) shows that our model was able to capture the anatomical structures of faces and generate content that is consistent with the holistic semantics.

**Attribute Controller** Unlike previous image completion techniques (Iizuka et al., 2017; Yeh et al., 2017; Li et al., 2017; Yang et al., 2016; Pathak et al., 2016; Yu et al., 2018) that generate only random plausible content, our network completes faces with structurally meaningful content whose appearances and expressions are controllable. Existing approaches (Mirza & Osindero, 2014; Choi et al., 2017; Kaneko et al., 2017) can only control facial expressions roughly (e.g. smiling or not smiling). In contrast, our model is able to control subtle expressions. In the experiment, the face appearance was conditioned on a "Male" attribute and we used landmarks from source actors to control the synthesized expressions (Figure 5). This $512 \times 512$ model was trained from scratch. The example demonstrates the potential application of our method in face reenactment (Li et al., 2012; Garrido et al., 2014; Thies et al., 2016).

**Computation Time** Our model, once trained, is able to complete a face image with a single forward pass, resulting in much higher efficiency. We tested our model with a Titan Xp GPU by processing 3000 $1024 \times 1024$ images with $512 \times 512$ holes. The mean completion time is 0.54 second per image. Unlike our model, existing CNN-based high-resolution in-painting approaches often need much longer time to process an image. For instance, it took about 1 min for the model of Yang et al. (Yang et al., 2016) to complete a $512 \times 512$ image with a Titan X GPU.

**User Study** We compared our method with CTX (Yu et al., 2018), which is the state-of-the-art CNN-based face completion approach capable of completing face images at $256 \times 256$ resolution, with a pilot user study at $256 \times 256$ resolution with random masks. 27 subjects (15 male and 12 female participants, with ages from 22 to 32) were volunteered to participate.

There were four sessions of pairwise A/B tests. Each time, a user was shown two images and asked choose the more realistic one. In the first session, two images completed from the same image by different methods were chosen. In session two to four, a real image and a corresponding synthesized image were shown. In the first session, time was unlimited. In session two to four, images were on display for 250ms, 1000ms, 4000ms respectively. The result (Figure 6) shows that there were significantly more images generated by our method being favored by the viewers. Overall, our approach generated sharper images with more details and fewer distortions.

**Limitations** Though our method has low inference time, the training time is long due to the progressive growing of networks. In our experiment, it took about three weeks to train a $1024 \times 1024$ model on a Titan Xp GPU. Additionally, by carefully zooming in our results, we find that our high-resolution model fails to learn low-level skin textures, such as furrows and sweat holes. Moreover, the model

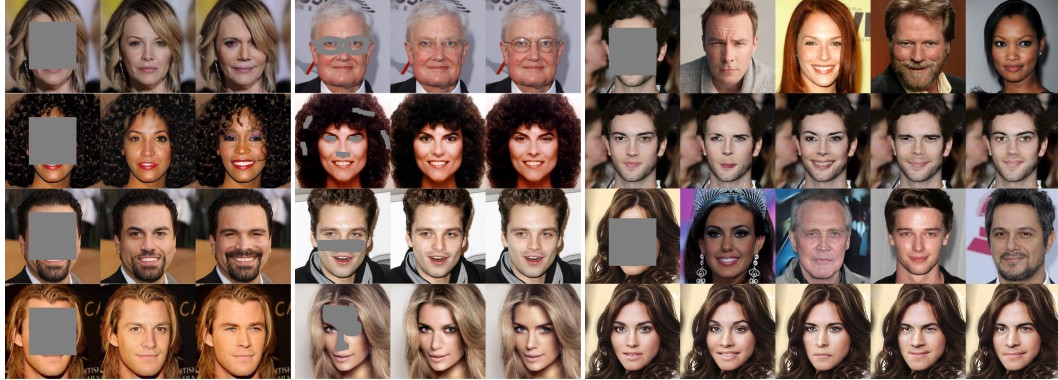

Figure 5: Sample results of our approach. The left two groups are completion results with center and irregular hand-drawn masks at $1024 \times 1024$. For each group, from left to right columns: cropped, synthesized and real images. The third group shows the performance of the attribute controller, in which the first and third row are corrupted images and source actors whose facial landmarks are used to control the expressions of synthesized faces (row two and four). The right most two columns are conditioned on the "Male" attribute while column two and three are with "Not Male". The leftmost column depends on their ground-truth landmarks and attributes.

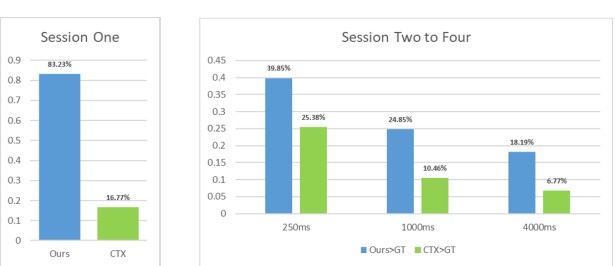
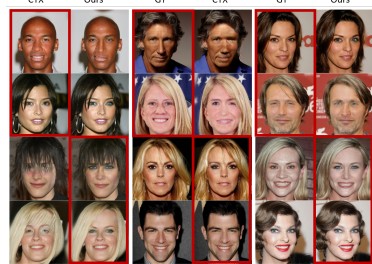

Figure 6: Comparisons on the naturalness: ours and CTX (Yu et al., 2018). The leftmost bar chart shows the average percentage that the images generated by our method look more natural than CTX. The second bar chart shows the percentage that a synthesized image is considered more realistic than a ground-truth (GT) one with displaying time of 250ms, 1000ms and 4000ms. The right figure shows samples used in the user study. The first group comes from session one while group two and three are both from session four (the 4000ms session). The preferred images are marked with red boxes.

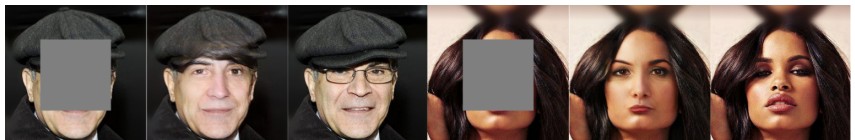

Figure 7: Some failure cases of our approach.

could generate distorted content while removing large parts (e.g. hats) or synthesize some plausible but unnatural faces (Figure 7). Furthermore, For facial expression transfer, our method requires that the head poses of the source and target faces are similar. These issues are left for future work.

## 5    CONCLUSION

We propose a progressive GAN with frequency-oriented attentive modules (FAM) for high-resolution and fast face completion, which learns face structures from coarse to fine guided by the FAM. By consolidating information across all scales, our model not only outperforms state-of-the-art methods by generating sharper images in low resolution, but is also able to synthesize faces in higher resolutions than existing techniques. A conditional version of our model allows users to control the properties of generated images explicitly with attribute vectors and landmarks. Our system is designed in an end-to-end manner, in that it learns to generate completed faces directly and more efficiently.

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

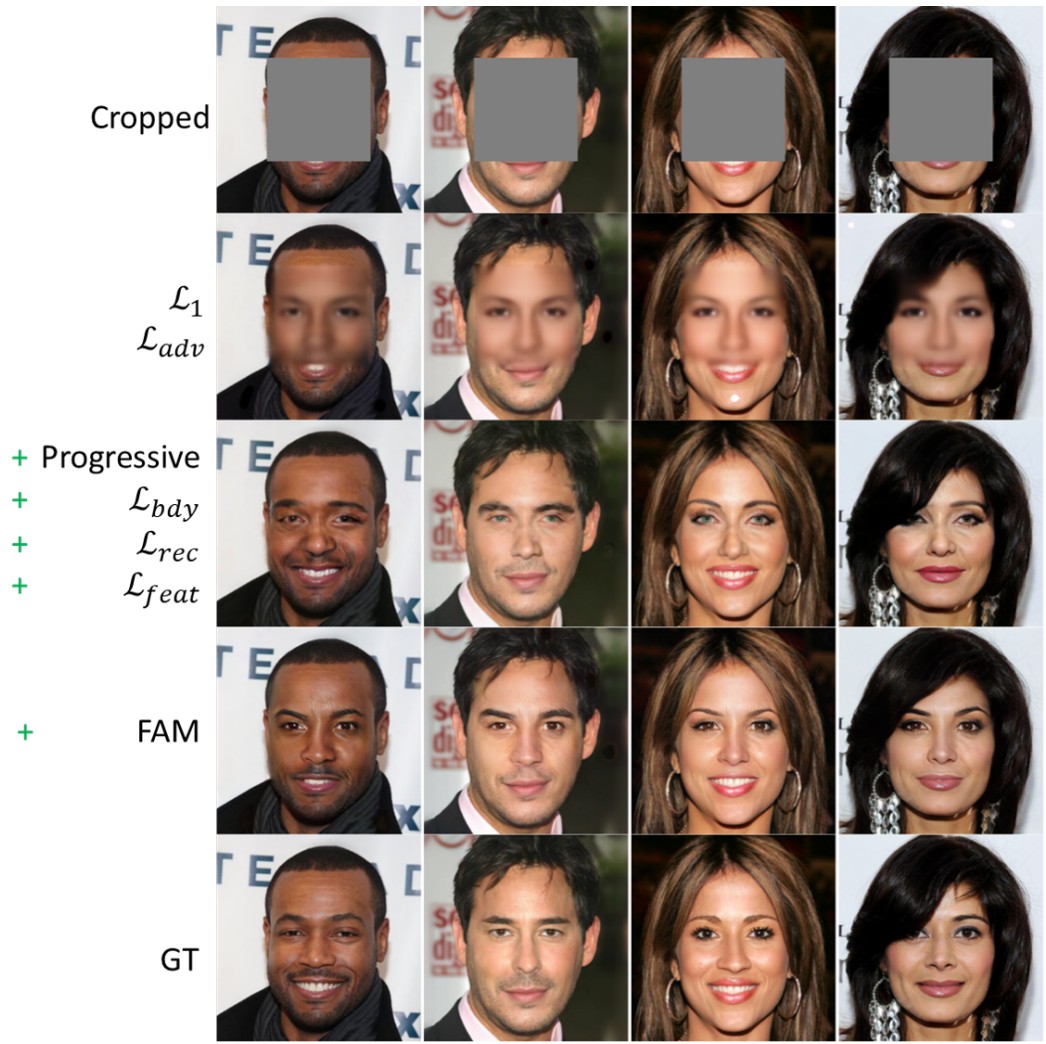

Figure 8: Ablation Study. First row: cropped images; Second row: results from a model that is trained with only regular $\mathcal{L}_1$ loss (unweighted) and $\mathcal{L}_{adv}$. The vanishing gradient problem prevents the networks from making progress when the synthesized content is still blurry. Third row: the training process is stabilized by adopting the progressive training methodology and a set of designed loss functions. However, the already-learned structures can be distorted while the network is growing (e.g. first column); Fourth row, FAM helps prevent the coarse structures from being altered while encouraging the model to attend to regions with rich details. For instance, the eyes are sharper, more vivid and realistic when the model is trained with FAM; Fifth row: the ground-truth samples.

# A APPENDIX

## A.1 ABLATION STUDY

The encoder-decoder structure has been widely used in image completion networks (Pathak et al., 2016; Iizuka et al., 2017; Li et al., 2017). However, the encoding process is a lossy compression, which makes it difficult to reconstruct the original contextual regions. Additionally, since much contextual information is lost during the encoding process, it is also difficult for the encoder to reconstruct content that perfectly match the context. UNet adds skip connections between the mirrored layers of the encoder and decoder, consolidating information from all previous layers, rather than depending on only the latent code. Therefore, UNet can be used to generate a completed image conditioned on the corrupted input. Unfortunately, if UNet is applied to image completion networks directly, it will be much easier for the generator to reconstruct the context than the content, resulting in inconsistent colors and textures along boundaries, which often causes the vanishing gradient problem

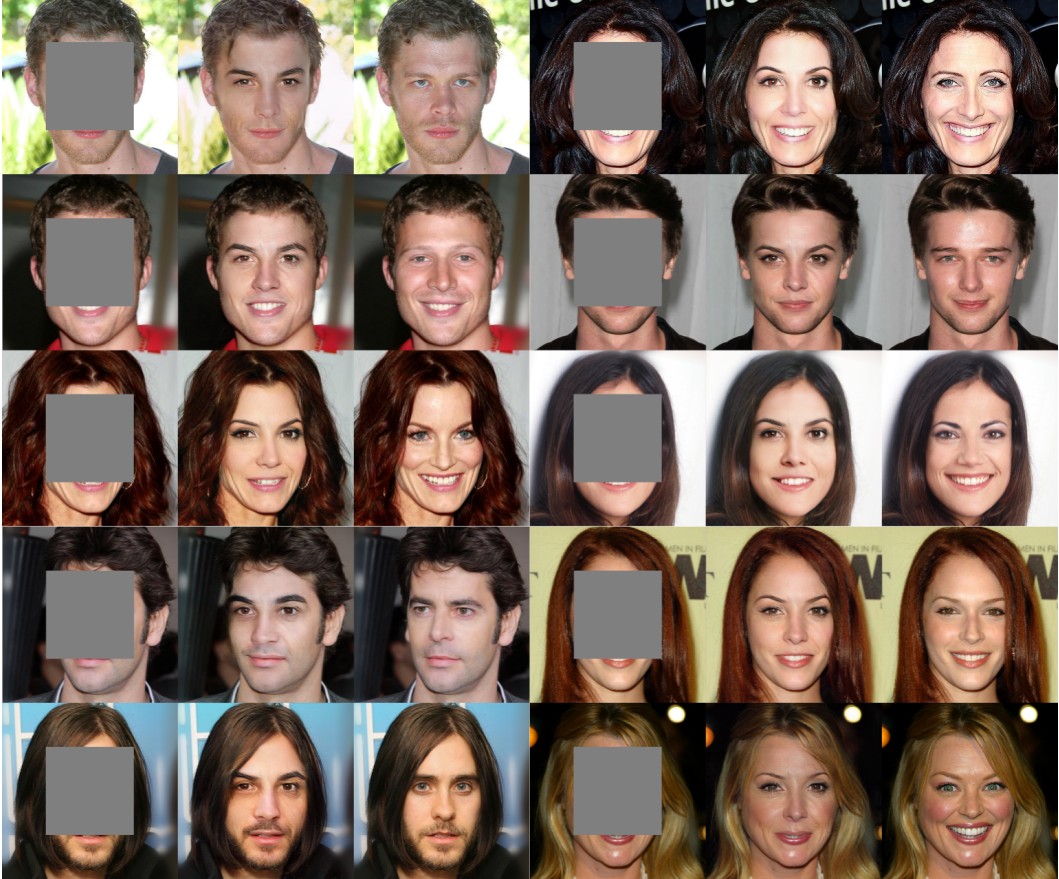

Figure 9: High-resolution face completion results with center masks. All images are resized from $1024 \times 1024$. For each group, from left to right: cropped, synthesized, and real images.

where both the generator and discriminator stop learning when the synthesized content is still blurry (Figure 8).

We try to solve this problem from two aspects. First, we adopt the method of training GANs progressively. Our model starts at a low resolution, so that it is difficult for the discriminator to capture the inconsistency between content and context regions. When the network grows, since the lower-resolution is already trained, the inconsistency between context and content regions is reduced. Second, we design the boundary loss and weighted reconstruction loss, which encourage the network focus more on synthesizing the boundary and content regions respectively. The feature loss also helps stabilizing the training by encouraging the synthesized images to have similar high-level features to real samples. These two improvements have improved the performance significantly.

The image quality is further enhanced by incorporating FAM. First, the reader and writer act as band-pass filters that minimize the influence of high-frequency noise on low-level network parameters, and thus avoiding distorting the already-learned global structures. Second, FAM is predicted from the facial semantics so that it encourages the model to focus on learning features in regions with richer details. For instance, the eyes synthesized by models trained with FAM are much sharper, more vivid and realistic.

This ablation study was run at $256 \times 256$ to provide an intuitive illustration of the impact of different components of our method. Since the training of high-resolution models was very time consuming, a more thorough ablation study is left for future work.

## A.2  HIGH-RESOLUTION COMPLETION RESULTS

More high resolution face completion results with various mask types are demonstrated in Figure 9 and Figure 10. Additionally, Figure 11 and Figure 12 represent two ways to control expressions

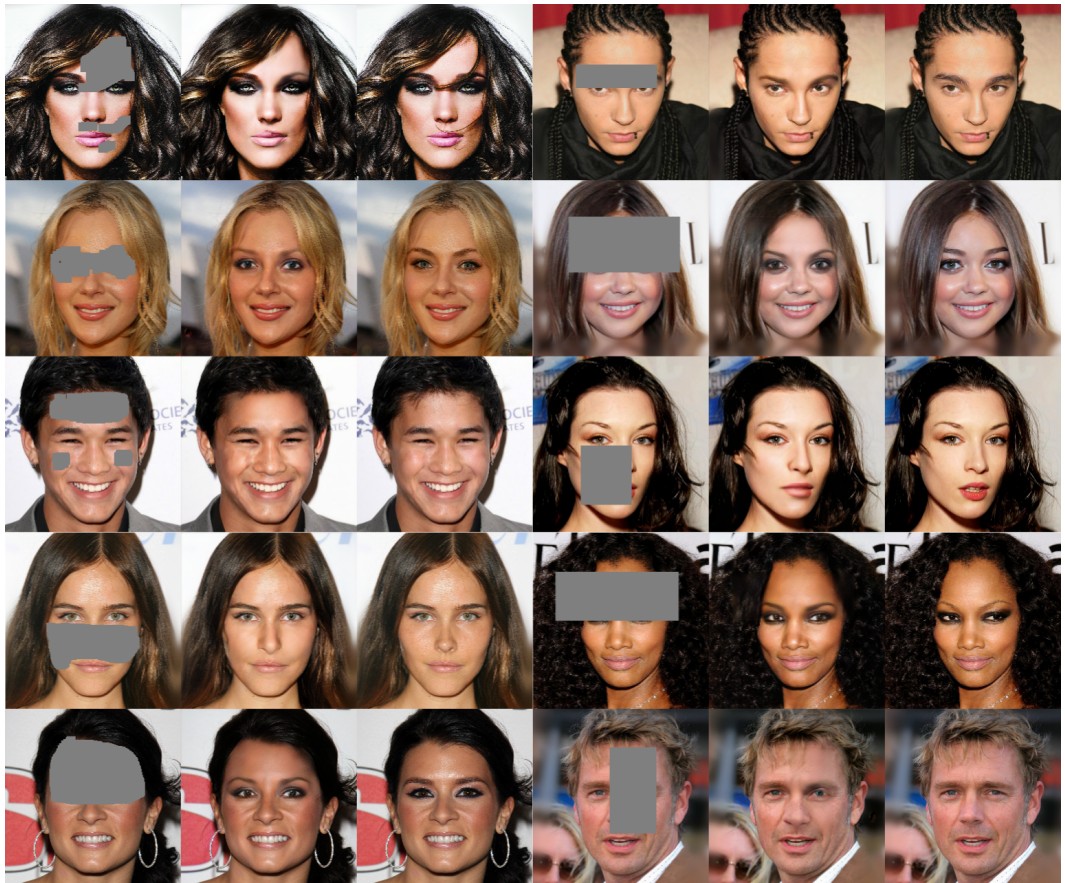

Figure 10: High-resolution face completion results with random and hand-drawn masks. All images are resized from $1024 \times 1024$. For each group, from left to right: cropped, synthesized, and real images.

and appearances. In the first example (Figure 11), we use a two-dimensional attribute vector ["Smiling", "Male"] without landmarks as inputs. In the second one, an attribute ["Male"] is used with landmarks extracted from source actors (Figure 12). The results show that both methods can control the expressions and appearances explicitly. But more subtle expressions can be controlled with landmarks. Moreover, we present more examples of attention filters during growing process in Figure 13.

### A.3 TRAINING DETAILS

The progressive training process is illustrated in Figure 14. At a resolution lower than $1024 \times 1024$, the input face images, masks, landmarks and real images are all down-sampled with average pooling to fit the given scale. One of the major challenges of generating high resolution images is the limitation of Graphics Processing Unit (GPU) memory. Most completion networks use Batch Normalization (Ioffe & Szegedy, 2015) to avoid covariate shift. However, with the limited GPU memory, only a small number of batch sizes are supported at high resolution, resulting in low quality of generated images. We use the Instance Normalization (Ulyanov et al., 2016), similar to Zhu et al. (Zhu et al., 2017), and update $D$ with a history of completed images instead of the latest generated one (Shrivastava et al., 2016) to stabilize training.

At the growing stage, new layers are added for both $D$ and $G$ and these layers are faded in with current networks smoothly. After the fade-in process, the network is trained on more images for stabilization. We used 300K, and 150K training images for resolution [$8 \times 8$ to $256 \times 256$] and [$512 \times 512$, $1024 \times 1024$] respectively at growing stage, and 600K, 430K images for $4 \times 4$ and [$8 \times 8$ to $1024 \times 1024$] at stabilizing stage respectively.

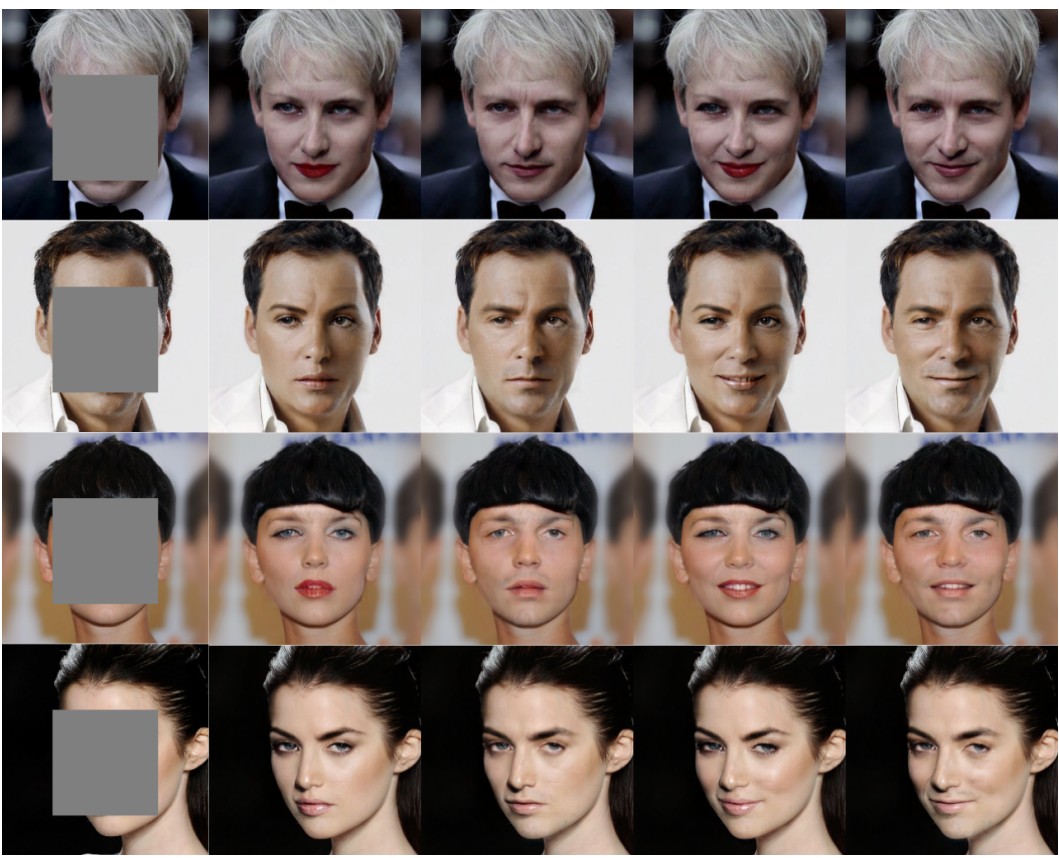

Figure 11: Face completion results with attribute controller. In this example, only attribute vectors (["Smiling", "Male"]) are used to control the properties of generated images. The facial expressions are controlled with the latent variables, rather than landmarks. From column two to five, the attributes are: [0, 0], [0, 1], [1, 0], [1, 1]. "1" denotes an attribute is turned on, otherwise not.

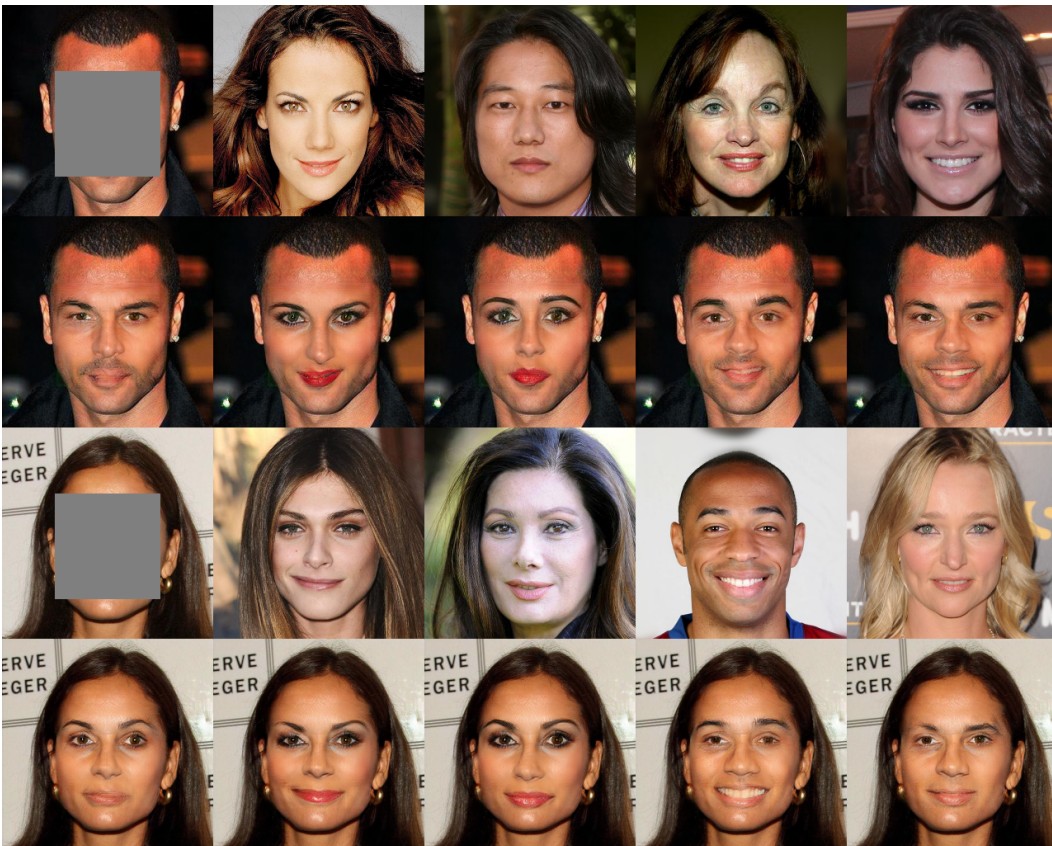

Figure 12: Face completion results with attribute controller. Attribute "Male" is used to control the appearances ("Male" for column two and three; "Not Male" for column four and five). Landmarks from source actors (row one and three) are used to control expressions of synthesized images (row tow and four). The leftmost column shows cropped images and faces generated with ground-truth attributes and landmarks.

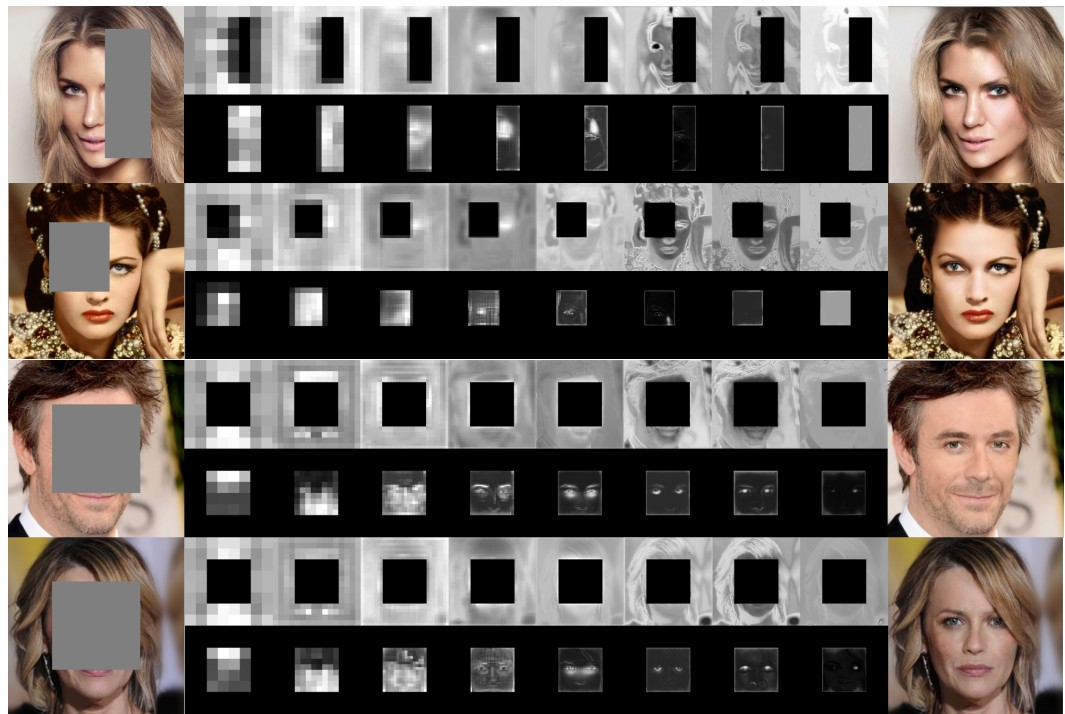

Figure 13: More examples of the attentive read/write filters while the resolution grows from $8 \times 8$ to $1024 \times 1024$. The leftmost column are cropped images while the rightmost are synthesized images.

In the experiments, the reconstruction trade-off parameter was set to $\kappa = 0.7$ to focus more on the target region. To balance the effects of different objective functions, we used $\lambda_{attr} = 2$, $\lambda_{rec} = 500$, $\lambda_{feat} = 8$, and $\lambda_{bdy} = 5000$. The Adam solver (Kingma & Ba, 2014) was employed with a learning rate of 0.0001.

## A.4 NETWORK STRUCTURE

The generator $G$ in our model is implemented by a U-shape network architecture consisting of the first component $G_{enc}$ transforming the observed image and its mask to a latent vector and the second component $G_{compl}$ transforming the concatenated vector (latent and attribute) to the completed image. There are residual connections between layers in $G_{enc}$ and the counterpart in $G_{compl}$ similar in the spirit to the U-Net (Ronneberger et al., 2015) and the Hourglass network (Newell et al., 2016) to consolidate information across multiple scales. Figure 15 illustrates the two structures of a layer in the generator for training without and with attributes respectively, which are adapted from the U-Net and Hourglass network.

Every convolutional layer (Conv) is followed by an Instance Normalization (InsNorm) and a LeakyReLU layer, except that the Conv before the latent vector (i.e. the second Conv layer in Table 2) is not followed by an InsNorm. Additionally, the there are no InsNorms or LeakyReLUs after the last Convs of both $D_{cls}$ and $D_{attr}$. All Convs used in the residual block of the skip connections of our conditional model have a kernel size of three and a stride of one.

Since we use Instance Normalization rather than Batch Normalization, the batch size is not an important hyper-parameter. Technically, for faster computation, we use as large a batch size as possible so long as it does not exceed the GPU memory limit.

Tables 2 and 4 demonstrate the architecture of the components of the generator $G$ while Tables 5 shows the components of the discriminator $D$. In Table 5, depending on the operation of the skip connection (Skip), the number of filters is either doubled (for a concatenation operation) or remains the same (for an addition operation).

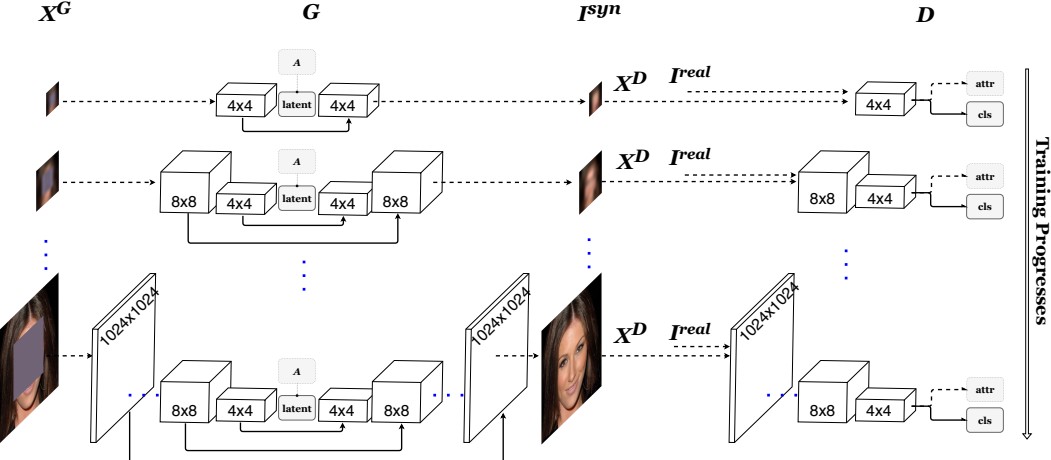

Figure 14: The progressive training process of our approach. The training of the completion network (or the "generator" $G$) and the discriminator $D$ starts at low resolution ($4 \times 4$). Higher layers are added to both $G$ and $D$ progressively to increase the resolution of the synthesized images. The $\boxed{r \text{ x } r}$ cubes in the figure represent convolutional layers that handle resolution $r$. For the conditional version, attribute labels $A^{obs}$ are concatenated to the latent vectors. The discriminator $D$ splits into two branches in the final layers: $D_{cls}$ that classifies if an input image is real, and $D_{attr}$ that predicts attribute vectors. Note that $X^G$ and $X^D$ are both a set of inputs as defined in the paper. We use images in this Figure as a simplified illustration.

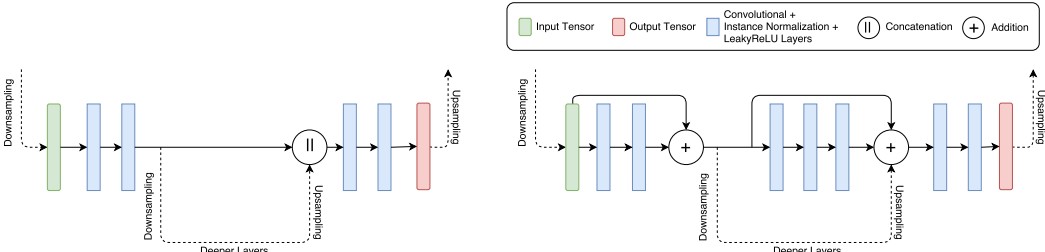

Figure 15: Illustrations of a single layer of our architecture. There are skip connections between mirrored encoder and decoder layers. Left: the structure of the completion network; the skip connection is a copy-and-concatenate operation. This structure helps preserve the identity information between the synthesized images and real faces, resulting in little deformation. Right: the structure of the conditional completion network; residual connections are added to the encoder, and the skip connections are residual blocks instead of direct concatenation. The attributes of the synthesized contents can be manipulated more easily with this structure. Each blue rectangle represents a set of Convolutional, Instance Normalization and Leaky Rectified Linear Unit (LeakyReLU) (Maas et al., 2013) layers.

Table 2: Top: the Encoding component of generator $G_{enc}$; Bottom: Latent Layer. $N$ is the length of an attribute vector. The attribute concatenation operation (AttrConcat) is only activated for our conditional model.

| Type | Kernel | Stride | Output Shape |
|------|--------|--------|--------------|
| Input Image | - | - | $4 \times 1024 \times 1024$ |
| Conv | $1 \times 1$ | $1 \times 1$ | $16 \times 1024 \times 1024$ |
| Conv | $3 \times 3$ | $1 \times 1$ | $32 \times 1024 \times 1024$ |
| Conv | $3 \times 3$ | $1 \times 1$ | $32 \times 1024 \times 1024$ |
| Downsample | - | - | $32 \times 512 \times 512$ |
| Conv | $3 \times 3$ | $1 \times 1$ | $64 \times 512 \times 512$ |
| Conv | $3 \times 3$ | $1 \times 1$ | $64 \times 512 \times 512$ |
| Downsample | - | - | $64 \times 256 \times 256$ |
| Conv | $3 \times 3$ | $1 \times 1$ | $128 \times 256 \times 256$ |
| Conv | $3 \times 3$ | $1 \times 1$ | $128 \times 256 \times 256$ |
| Downsample | - | - | $128 \times 128 \times 128$ |
| Conv | $3 \times 3$ | $1 \times 1$ | $256 \times 128 \times 128$ |
| Conv | $3 \times 3$ | $1 \times 1$ | $256 \times 128 \times 128$ |
| Downsample | - | - | $256 \times 64 \times 64$ |
| Conv | $3 \times 3$ | $1 \times 1$ | $512 \times 64 \times 64$ |
| Conv | $3 \times 3$ | $1 \times 1$ | $512 \times 64 \times 64$ |
| Downsample | - | - | $512 \times 32 \times 32$ |
| Conv | $3 \times 3$ | $1 \times 1$ | $512 \times 32 \times 32$ |
| Conv | $3 \times 3$ | $1 \times 1$ | $512 \times 32 \times 32$ |
| Downsample | - | - | $512 \times 16 \times 16$ |
| Conv | $3 \times 3$ | $1 \times 1$ | $512 \times 16 \times 16$ |
| Conv | $3 \times 3$ | $1 \times 1$ | $512 \times 16 \times 16$ |
| Downsample | - | - | $512 \times 8 \times 8$ |
| Conv | $3 \times 3$ | $1 \times 1$ | $512 \times 8 \times 8$ |
| Conv | $3 \times 3$ | $1 \times 1$ | $512 \times 8 \times 8$ |
| Downsample | - | - | $512 \times 4 \times 4$ |

| Type | Kernel | Stride | Output Shape |
|------|--------|--------|--------------|
| Conv | $3 \times 3$ | $1 \times 1$ | $512 \times 4 \times 4$ |
| Conv | $4 \times 4$ | $1 \times 1$ | $512 \times 1 \times 1$ |
| AttrConcat | optional | - | $512(+N) \times 1 \times 1$ |
| Conv | $4 \times 4$ | $1 \times 1$ | $512 \times 4 \times 4$ |
| Conv | $3 \times 3$ | $1 \times 1$ | $512 \times 4 \times 4$ |

Table 3: The completion component of generator $G_{compl}$. Depending on the particular operation of the skip connection (Skip), the number of filters is either doubled (for concatenation operations) or remains the same (for addition operations). In practice, $G_{compl}$ output a feature map that can be used to generate a RGB image (with *ToRGB* layers) or predict a read/write Filter (with *ToFilter* layers, see Table 4).

| Type | Kernel | Stride | Output Shape |
|------|--------|--------|--------------|
| Upsample | - | - | $512 \times 8 \times 8$ |
| Skip | - | - | $1024\,(512) \times 8 \times 8$ |
| Conv | $3 \times 3$ | $1 \times 1$ | $512 \times 8 \times 8$ |
| Conv | $3 \times 3$ | $1 \times 1$ | $512 \times 8 \times 8$ |
| Upsample | - | - | $512 \times 16 \times 16$ |
| Skip | - | - | $1024\,(512) \times 16 \times 16$ |
| Conv | $3 \times 3$ | $1 \times 1$ | $512 \times 16 \times 16$ |
| Conv | $3 \times 3$ | $1 \times 1$ | $512 \times 16 \times 16$ |
| Upsample | - | - | $512 \times 32 \times 32$ |
| Skip | - | - | $1024\,(512) \times 32 \times 32$ |
| Conv | $3 \times 3$ | $1 \times 1$ | $512 \times 32 \times 32$ |
| Conv | $3 \times 3$ | $1 \times 1$ | $512 \times 32 \times 32$ |
| Upsample | - | - | $512 \times 64 \times 64$ |
| Skip | - | - | $1024\,(512) \times 64 \times 64$ |
| Conv | $3 \times 3$ | $1 \times 1$ | $512 \times 64 \times 64$ |
| Conv | $3 \times 3$ | $1 \times 1$ | $512 \times 64 \times 64$ |
| Upsample | - | - | $512 \times 128 \times 128$ |
| Conv | $3 \times 3$ | $1 \times 1$ | $256 \times 128 \times 128$ |
| Skip | - | - | $512\,(256) \times 128 \times 128$ |
| Conv | $3 \times 3$ | $1 \times 1$ | $256 \times 128 \times 128$ |
| Conv | $3 \times 3$ | $1 \times 1$ | $256 \times 128 \times 128$ |
| Upsample | - | - | $256 \times 256 \times 256$ |
| Conv | $3 \times 3$ | $1 \times 1$ | $128 \times 256 \times 256$ |
| Skip | - | - | $256\,(128) \times 256 \times 256$ |
| Conv | $3 \times 3$ | $1 \times 1$ | $128 \times 256 \times 256$ |
| Conv | $3 \times 3$ | $1 \times 1$ | $128 \times 256 \times 256$ |
| Upsample | - | - | $128 \times 512 \times 512$ |
| Conv | $3 \times 3$ | $1 \times 1$ | $64 \times 512 \times 512$ |
| Skip | - | - | $128\,(64) \times 512 \times 512$ |
| Conv | $3 \times 3$ | $1 \times 1$ | $64 \times 512 \times 512$ |
| Conv | $3 \times 3$ | $1 \times 1$ | $64 \times 512 \times 512$ |
| Upsample | - | - | $64 \times 1024 \times 1024$ |
| Conv | $3 \times 3$ | $1 \times 1$ | $32 \times 1024 \times 1024$ |
| Skip | - | - | $64\,(32) \times 1024 \times 1024$ |

Table 4: Left: The *ToRGB* layers that convert feature maps to RGB images. Right: *ToFilter* layers that predict a read/write filter from feature maps.

| Conv | $3 \times 3$ | $1 \times 1$ | $32 \times 1024 \times 1024$ |
|------|--------------|--------------|------------------------------|
| Conv | $3 \times 3$ | $1 \times 1$ | $32 \times 1024 \times 1024$ |
| Conv | $1 \times 1$ | $1 \times 1$ | $3 \times 1024 \times 1024$ |

| Conv | $3 \times 3$ | $1 \times 1$ | $64 \times 1024 \times 1024$ |
|------|--------------|--------------|------------------------------|
| Conv | $3 \times 3$ | $1 \times 1$ | $64 \times 1024 \times 1024$ |
| Conv | $1 \times 1$ | $1 \times 1$ | $1 \times 1024 \times 1024$ |

Table 5: Top: Feature Network $\mathbb{F}(\cdot)$ computes a feature map for an input image, which is later used by $D_{cls}$ and $D_{attr}$; Middle: The real/fake head classifier $D_{cls}$; Bottom: The attribute network $D_{attr}$. $N$ is the length of an attribute vector. This network is only activated for the conditional model.

| Type | Kernel | Stride | Output Shape |
|---|---|---|---|
| Input Image | - | - | $3 \times 1024 \times 1024$ |
| Conv | $1 \times 1$ | $1 \times 1$ | $16 \times 1024 \times 1024$ |
| Conv | $3 \times 3$ | $1 \times 1$ | $16 \times 1024 \times 1024$ |
| Conv | $3 \times 3$ | $1 \times 1$ | $32 \times 1024 \times 1024$ |
| Downsample | - | - | $32 \times 512 \times 512$ |
| Conv | $3 \times 3$ | $1 \times 1$ | $32 \times 512 \times 512$ |
| Conv | $3 \times 3$ | $1 \times 1$ | $64 \times 512 \times 512$ |
| Downsample | - | - | $64 \times 256 \times 256$ |
| Conv | $3 \times 3$ | $1 \times 1$ | $64 \times 256 \times 256$ |
| Conv | $3 \times 3$ | $1 \times 1$ | $128 \times 256 \times 256$ |
| Downsample | - | - | $128 \times 128 \times 128$ |
| Conv | $3 \times 3$ | $1 \times 1$ | $128 \times 128 \times 128$ |
| Conv | $3 \times 3$ | $1 \times 1$ | $256 \times 128 \times 128$ |
| Downsample | - | - | $256 \times 64 \times 64$ |
| Conv | $3 \times 3$ | $1 \times 1$ | $256 \times 64 \times 64$ |
| Conv | $3 \times 3$ | $1 \times 1$ | $512 \times 64 \times 64$ |
| Downsample | - | - | $512 \times 32 \times 32$ |
| Conv | $3 \times 3$ | $1 \times 1$ | $512 \times 32 \times 32$ |
| Conv | $3 \times 3$ | $1 \times 1$ | $512 \times 32 \times 32$ |
| Downsample | - | - | $512 \times 16 \times 16$ |
| Conv | $3 \times 3$ | $1 \times 1$ | $512 \times 16 \times 16$ |
| Conv | $3 \times 3$ | $1 \times 1$ | $512 \times 16 \times 16$ |
| Downsample | - | - | $512 \times 8 \times 8$ |
| Conv | $3 \times 3$ | $1 \times 1$ | $512 \times 8 \times 8$ |
| Conv | $3 \times 3$ | $1 \times 1$ | $512 \times 8 \times 8$ |
| Downsample | - | - | $512 \times 4 \times 4$ |

| Type | Kernel | Stride | Output Shape |
|---|---|---|---|
| Conv | $3 \times 3$ | $1 \times 1$ | $512 \times 4 \times 4$ |
| Conv | $4 \times 4$ | $1 \times 1$ | $1 \times 1 \times 1$ |

| Type | Kernel | Stride | Output Shape |
|---|---|---|---|
| Conv | $3 \times 3$ | $1 \times 1$ | $512 \times 4 \times 4$ |
| Conv | $4 \times 4$ | $1 \times 1$ | $N \times 1 \times 1$ |

