# OpenReview forum: "High Resolution and Fast Face Completion via Progressively Attentive GANs"
_ICLR.cc/2019/Conference_

### Official Review · AnonReviewer2 · 2018-10-22
**High quality results but limited novelty. Need more evidence of improvements over previous methods**

**Rating:** 5
**Confidence:** 5

**Review:**

This paper proposed a new method for face completion using progressive GANs. The novelty seems very limited compared with previous methods. The results did not significantly outperform previous methods such as CTX in terms of visual quality. In addition, some of the features for the proposed method were not evaluated properly.

1. The frequency attention module is not convincing. The visualization of the attention features look like normal feature in a neural network. Also, in Figure 8, the quality of results with and without FAM look very similar. These 4 images were selected from 3000 test images, but the difference is too small to show the benefit of FAM.

2. In figure 8, it is unclear how the performance changes with each loss. Probably the results without L_bdy, L_rec, L_feat should be analyzed separately.

3.  In figure 6, the results compared to CTX look similar. And the figure is too small to see the details. For example, from row 1, the result by CTX seems even better.

4. How many images were used in the user study? Did each subjects evaluate the entire test set 3009 images?

---

> ### Author Response · Authors · 2018-11-17
> **Response to Reviewer Three**
>
> “The results did not significantly outperform previous methods such as CTX in terms of visual quality…In figure 6, the results compared to CTX look similar. And the figure is too small to see the details.”
>
> When we chose the sample images (Figure 6), we did not intentionally choose bad samples of CTX and good samples of our approach. Instead, we want to demonstrate some typical cases when each of these approaches failed or succeeded. Since it is not fair to compare the performance of two GAN approaches by looking at only a few samples, we used the results of a user study, which is known as the “golden standard” to evaluate GANs, to show the overall performance of different approaches, which we think should be more convincing.
>
> “The visualization of the attention features look like normal feature in a neural network”
>
> The filters (Figure 1) showed clear and regular patterns as we expected. For instance, while the resolution increased from 8x8 to 1024x1024, the model attended on higher frequency information. Regions with rich details (e.g. eyes) got more attention, especially at high resolutions. It is unlikely that they are simply some normal features in a neural network.
>
> “in Figure 8, the quality of results with and without FAM look very similar. These 4 images were selected from 3000 test images, but the difference is too small to show the benefit of FAM.”
>
> The FAM is designed to enhance details. If we look closely at the third and fourth row of Figure 8, the results without FAM are blurrier, especially at regions with rich details (e.g. eye regions). Also, results with FAM usually have less distortions.
>
> Again, we demonstrated the typical performances of models with and without FAM, instead of intentionally choosing images that showed the worst cases of images without FAM.
>
> “In figure 8, it is unclear how the performance changes with each loss. Probably the results without L_bdy, L_rec, L_feat should be analyzed separately.”
>
> This is a good suggestion, it would be better to do a more thorough ablation study.  However, the effects of many losses (e.g. L_rec, L_feat) have been well studied in previous literatures (e.g. Li et al., 2017) and thus they are not the focus of our work.
>
> “How many images were used in the user study? Did each subjects evaluate the entire test set 3009 images?”
>
> For session 1 where the experiment directly compares our method with context encoder, each subject evaluates 100 chosen randomly images. For session 2, 3 and 4 where each method compares with ground truth, each subject evaluates another random 100 images. The total coverage rate over the entire test set is about 86%.

---

> > ### Comment · AnonReviewer2 · 2018-11-27
> > **The rebuttal is not convincing**
> >
> > The authors claimed that the results in the paper are some "typical cases", neither completely random, nor cherry-picked bad results for CTX. However, "typical cases" are still very vague. The author still did some kind of selection. Since there are user studies, why not showing some top selected results and least selected results by the users when comparing with CTX. It is weird to say that one method is much better than the other, while the images look very similar.
> >
> > The response is also self-contradictory. The author stated that "it is not fair to compare the performance of two GAN approaches by looking at only a few samples", but only a few examples are selected to demonstrate the benefit of FAM in Fig. 8. Since FAM is one of the claimed novel parts of the paper, why not including the results without FAM in the user study.

---

### Official Review · AnonReviewer1 · 2018-10-23

**Rating:** 5
**Confidence:** 2

**Review:**

The paper proposes a complex generative framework for image completion (particularly human face completion). It aims at solving the following challenges: 1) complete the human face at both low and high resolution; 2) control the attribute of the synthetic content; 3) without the need of complex post-processing. To achieve so, this paper proposes a progressively attentive GAN to complete face image at high resolution with multiple controllable attributes in a single forward pass without post-processing. Particularly it introduces a frequency-oriented attentive module (FAM) to attend on finer details.

The method seems interesting, however it seems to make slight change based on ProGAN (ICLR' 18   https://arxiv.org/abs/1710.10196). Also similar idea could be found in many other papers, e.g., Wang et al. High-Resolution Image Synthesis and Semantic Manipulation with Conditional GANs, CVPR' 18.

The authors should
1) clarify why this paper makes non-incremental contribution? What are the major novelty compared with these existing works?
2) why the frequency attention module will yield better results?
3) Improve the experiment, compared with stronger baselines: consider at least one or two of these state-of-the-art approaches. Also in my opinion model size and training time needs to be compared as well.

Also the experimental results did not demonstrate better performance of the proposed approach. Why is that?

---

> ### Author Response · Authors · 2018-11-17
> **Response to Reviewer Two**
>
> Thanks for the professional reviews. We would like to make some clarification to better demonstrate our work.
>
> “Also the experimental results did not demonstrate better performance of the proposed approach. Why is that?”
>
> Could you please explain which part of the results you are referring to?
> Both the results of the quantitative evaluation and user study showed our model performed better. In Table 1, for L1 and L2, the smaller value is better. For PSNR, the larger value is better. In Figure 6, the larger value is better.
>
> “What are the major novelty compared with these existing works? (Progressive GAN and Wang et al.)”
>
> The Progressive GAN is an image GENERATION network and the work of Wang et al. is an image TRANSLATION network. They are both different from the image COMPLETION networks (e.g. Pathak et al., 2016, Li et al., 2017, Iizuka et al., 2017, Yu et al., 2018, Liu et al., 2018, etc.) in terms of goals, network structures, training methods and loss functions, and are not directly comparable with our model. Neither of the Progressive GAN nor the work of Wang et al. can be applied to the image completion task directly, though some of their can be adopted to design completion models (e.g. the progressive training methodology in Progressive GAN).
>
> The input of an image generation model (e.g. Progressive GAN) is noise, and the output is a random realistic image. The image completion task is more challenging because it not only requires generating plausible content, but also expects the generated content to match the contextual information perfectly.
>
> The input of an image translation model is a complete image from one domain (e.g. segmentation labels), and the output is a transformed image in another domain, such as a realistic photo or a painting of another style (e.g. Zhu et al., 2017). The key difference is that some information is missing in the input of an image completion network, and the completion model needs to infer plausible content conditioned on contextual information.
>
> Therefore, it is more reasonable to compare our work with other completion models, rather than a generation or translation model. As we discussed in the response to R1, we have adopted many ideas from networks outside the image completion area and successfully integrate them to obtain an effective completion model. We have also designed novel structures, pipelines and loss functions so that our model can work appropriately as a whole. To our knowledge, our method is quite unique comparing to other image completion networks.
>
> “why the frequency attention module will yield better results?”
>
> Traditionally, researchers use the attention mechanism in spatial domain. Instead of learning to generate/complete the whole image at once, the model is encouraged to focus on a small region in one step. For instance, the DRAW model (Gregor et al., 2015) learns to read and write a small region of image in each timestep, and the whole image can be produced after many iterations. CTX (Yu et al., 2018) uses a contextual attention layer to help the model borrow contextual information from distant locations while filling in missing “holes”.
>
> Like these spatial-attention-based methods, we design an attention mechanism in frequency domain. Instead of generating image features at different level of details in a single step, our model is encouraged to learn the structures in a coarse-to-fine manner. The detailed design of FAM is described in Section 3.2.1. The results (Figure 1) shows that our model performed as we expected: it focused on coarse structures when the resolution was low and switched its attention to finer details (e.g. hair or eye regions) as the resolution increased. This attention mechanism works because the complex problem of completing high-resolution images is divided into many sub-problems.
>
> “Improve the experiment, compared with stronger baselines: consider at least one or two of these state-of-the-art approaches”
>
> CTX (Yu et al., cvpr18) is considered state-of-the-art. When we ran the user study and it was the only approach that worked for 256x256 images. We also included the comparison with another state-of-the-art approach GL (Iizuka et al., siggraph17) in the quantitative comparison (Table 1).

---

### Official Review · AnonReviewer3 · 2018-11-05
**This work uses GANs to recover clean faces from occluded counterparts. The effectiveness of the proposed method is verified qualitatively and quantitatively on CelebA-HQ. The proposed framework can be generalized to several face-related tasks, such as unconstrained face recognition. Although the novelty of the method is not really impressive, the proposed method seems to be useful for face-related applications and the experimental results are convincing to me.**

**Rating:** 5
**Confidence:** 5

**Review:**

This work uses GANs to recover clean faces from occluded counterparts. The effectiveness of the proposed method is verified qualitatively and quantitatively on CelebA-HQ. The proposed framework can be generalized to several face-related tasks, such as unconstrained face recognition. Although the novelty of the method is not really impressive, the proposed method seems to be useful for face-related applications and the experimental results are convincing to me.

Pros:
- This method is simple, apparently effective and is a nice use of GANs for a practical task. The paper is written clearly and the English is fine.

Cons:
- My main concern with this paper is regarding the novelty. The authors seem to claim a novel GAN architecture by using an adversarial auto-encoder-based architecture. However, it is not clear to me what aspect of their GAN is particularly new.

- Missing experimental comparisons with state-of-the-arts. Detailed experimental comparisons with more state-of-the-arts (e.g., RLA, Zhao et al., TIP 2018, 3D-PIM, Zhao et al., IJCAI 2018) are needed to justify the superiority of the proposed method.

- Missing more in-the-wild comparisons in the Experiment section. This paper mainly performed experiments on CelebA-HQ. More in-the-wild qualitative and quantitative experiments on recent benchmarks with large occlusion variations are needed to verify the efficacy of the proposed method.

Additional comments:
- How did authors update each component and ensure stable yet fast convergence while optimising the whole GAN-based framework?

- Can the proposed method solve other challenging in-the-wild facial variations except occlusion? e.g., pose, expression, lighting, noise, etc.

---

> ### Author Response · Authors · 2018-11-17
> **Response to Reviewer One**
>
> Thanks for the professional reviews. We would like to make some clarification to better demonstrate our work.
>
> “it is not clear to me what aspect of their GAN is particularly new”
>
> We agree that some building blocks of our model, such as the Context Encoder structure (Pathak et al., 2016), Progressive Training Methodology (Progressive GAN, Karras et al., 2017), Conditional GAN (Mirza et al., 2014) etc., are based on existing approaches. But it is a challenging task to integrate these methods to obtain an effective completion model. On the top of these existing approaches, we have also designed new structures (e.g. our novel Frequency-Oriented Attentive Module), novel pipeline (Figure 2) and loss functions (e.g. boundary loss) to significantly improve the performance.
>
> Please note that most of these building blocks are not originally designed for image completion and are seldomly used in completion models. For instance, the Progressive GAN is an image GENERATION model whose input is noise and the output are random realistic images. However, image COMPLETION is a more challenging task. Conditioned on corrupted images (i.e. the input), we not only need to generate plausible content, but also need to make sure that the content matches the contextual information perfectly. In sum, our network structure is significantly different from any of the existing approaches we built on. Additionally, to our knowledge, our method is also unique in the image completion area.
>
> Because of the novel architecture/method, our model achieves significantly better performance than state-of-the-art approaches. First, our model is the first one that can complete face images at 1024x1024 while state-of-the-art (CTX, Yu et al., 2018) can only handle 256x256 images. By running a user study, which is currently the “golden standard” to evaluate GANs, we found our model outperformed CTX in terms of visual quality at 256x256 resolution. Second, our model can control multiple attributes of synthesized content (including subtle facial expressions) while other completion models can only produce random content images. Third, our model does not need post-processing and can generate completed images directly while other approaches often have to post-process images (e.g. Lizuka et al., 2017) or paste synthesized content to original context (e.g. Yu et al., 2018).
>
> “Detailed experimental comparisons with more state-of-the-arts (e.g., RLA, Zhao et al., TIP 2018, 3D-PIM, Zhao et al., IJCAI 2018) are needed to justify the superiority of the proposed method”
>
> Thanks. We will include these literatures in our reference and compare with them in the future experiments.
>
> “More in-the-wild qualitative and quantitative experiments on recent benchmarks with large occlusion variations are needed to verify the efficacy of the proposed method.”
>
> Agreed, this is a good suggestion. But our current experiments followed the standard of experiments in state-of-the-art works (Pathak et al., 2016, e.g. Iizuka et al., 2017, Yu et al., 2018, etc.) and tested the performance of our model for various challenging mask types including center squared, random rectangular and arbitrary hand-drawn masks.
>
> “How did authors update each component and ensure stable yet fast convergence while optimizing the whole GAN-based framework?”
>
> We started with empirical parameters of existing approaches and updated them with trial and error .
>
> “Can the proposed method solve other challenging in-the-wild facial variations except occlusion? e.g., pose, expression, lighting, noise, etc.”
>
> This is an interesting idea. We focused on solving the face completion (or the “inpainting”) problem in this paper. But it would be great if we could apply our model to other tasks. This is left for our future work.

---

### Meta-Review · Area_Chair1 · 2018-12-13
**All reviewers assess the paper as being marginally below acceptance threshold**

**Confidence:** 4
**Recommendation:** Reject

**Metareview:**

All reviewers gave a 5 rating.
The author rebuttal was not able to alter the consensus view of reviewers.
See below for details.